

# Comparison of pathological characteristics between self-detected and screen-detected invasive breast cancers in Chinese women: a retrospective study

Qi Zhang[1],[*], Lanjun Ding[2],[3],[*], Xuan Liang[2],[3], Yuan Wang[2],[3], Jiao Jiao[1], Wenli Lu[2],[3] and Xiaojing Guo[1]

[1] Department of Breast Pathology and Lab, Tianjin Medical University Cancer Institute and Hospital, Tianjin, China
[2] Department of Epidemiology and Health Statistics, Tianjin Medical University, Tianjin, China
[3] Collaborative Innovation Center of Chronic Disease Prevention and Control, Tianjin Medical University, Tianjin, China
[*] These authors contributed equally to this work.

## ABSTRACT

**Background:** In China, there is insufficient evidence to support that screening programs can detect breast cancer earlier and improve outcomes compared with patient self-reporting. Therefore, we compared the pathological characteristics at diagnosis between self-detected and screen-detected cases of invasive breast cancer at our institution and determined whether these characteristics were different after the program's introduction (vs. prior to).

**Methods:** Three databases were selected (breast cancer diagnosed in 1995–2000, 2010, and 2015), which provided a total of 3,014 female patients with invasive breast cancer. The cases were divided into self-detected and screen-detected groups. The pathological characteristics were compared between the two groups and multiple imputation and complete randomized imputation were used to deal with missing data.

**Results:** Compared with patient self-reporting, screening was associated with the following factors: a higher percentage of stage T1 tumors (75.0% vs 17.1%, $P = 0.109$ in 1995–2000; 66.7% vs 40.4%, $P < 0.001$ in 2010; 67.8% vs 35.7%, $P < 0.001$ in 2015); a higher percentage of tumors with stage N0 lymph node status (67.3% vs. 48.4%, $P = 0.007$ in 2010); and a higher percentage of histologic grade I tumors (22.9% vs 13.9%, $P = 0.017$ in 2010).

**Conclusion:** Screen-detected breast cancer was associated with a greater number of favorable pathological characteristics. However, although screening had a beneficial role in early detection in China, we found fewer patients were detected by screening in this study compared with those in Western and Asian developed countries.

Corresponding authors
Wenli Lu, luwenli@tmu.edu.cn
Xiaojing Guo,
guoxiaojing@tjmuch.com

## INTRODUCTION

Breast cancer has become the major cause of death in Chinese women (*Li, Mello-Thoms & Brennan, 2016*; *Yang et al., 2005*). According to Chinese urban cancer registries, the overall

incidence of breast cancer has increased at a rate of 2–5% annually, with a peak incidence at an age of approximately 50 years (*Anderson et al., 2008*; *Porter, 2008*; *Song et al., 2015*; *Yang et al., 2005*). Early tumor detection, before symptoms appear, could significantly improve survival (*Tabar et al., 2001*; *Duffy et al., 2002*; *de Gelder et al., 2015*; *Sankatsing et al., 2017*).

The National Health and Family Planning Commission of the People's Republic of China organized a three-year breast cancer screening program for women aged 35–69 years between 2009 and 2011, with a second phase of screening launched in 2012 (*Song et al., 2015*). The first phase of the program screened 1.2 million women and detected 440 cases with early-stage lesions, giving a diagnostic rate of 48.0 per 100,000 women (*Song et al., 2015*). Concurrently, in 2009, the All-China Women's Federation and the National Health and Family Planning Commission organized a screening program that offered free screening for breast and cervical cancer to women in rural China. As of 2014, about 48.35 million women in rural China had received free tests since the program's inception. The guidelines for breast cancer screening in China, which were first published in 2007 and updated in 2015, recommend women at average risk of breast cancer are encouraged to have mammography combined with clinical breast examination after age 40 years. Even though, there is no national organized screening program (*Chinese Anticancer Association Breast Cancer Society, 2015*). *Zhu et al. (2014)* reported the long-term prognosis of breast cancers has been improved during the past 40 years. This article was aimed to observe whether the distribution of pathological characteristics at diagnosis had differed since the introduction of limited screening programs.

Studies worldwide have indicated that screen-detected patients have more favorable survival outcomes compared with the patients with self-discovered breast cancer (i.e., self-detected cancer) (*Wang, Tan & Chow, 2011*; *Joensuu et al., 2004*; *Kim et al., 2012*). Screen-detected cancers tend to be of a smaller size, to have better differentiation, and to be at an earlier stage (*Crispo et al., 2013*). In a study carried out in a private hospital in Hong Kong, patients with screen-detected breast cancer had greater numbers of favorable pathological characteristics than a self-detected group (*Lau et al., 2016*). Therefore, the second aim of this study was to compare the pathological characteristics of the self-detected (symptomatic) and screen-detected (asymptomatic) invasive breast cancer in Tianjin, China.

## MATERIALS AND METHODS

### Information of database and subjects

This was a retrospective cohort study conducted at the Tianjin Medical University Cancer Institute and Hospital. Since 1995, all cases of breast cancer treated in this hospital have been recorded in a structured database. We identified cases for 1995–2000 (paper documentation), 2010 (half paper and half electronic documentation), and 2015 (electronic documentation), taking care to exclude those cases with ductal carcinoma in situ and bilateral breast cancer. The study was approved by the Ethics Committee in Tianjin Medical University Cancer Institute and Hospital.

## Data extraction

Clinical histories and pathological characteristics were obtained from the three databases by two authors individually (Q. Zhang and L. Ding), including the age of patients at initial diagnosis and the cancer detection method. Different records between authors were re-checked. Pathological characteristics included tumor size staging and lymph node staging and histologic grade based, respectively, on the tumor-node-metastasis classification system of the American Joint Committee on Cancer (*Edge & Compton, 2010*) and the World Health Organization classification of tumors (*Lakhani et al., 2012*).

## Methods of detection

Cases were divided into two groups, based on method of detection: a self-detected group and a screen-detected group. Patients in the screen-detected group were primarily screened by population-based or opportunistic screening with mammography, ultrasound, or clinical breast examination. Patients in the self-detected group were defined as those with obvious clinical symptoms at presentation, including nipple discharge, pain, a palpable axillary lump, a palpable breast lump, or a combination of those symptoms.

## Statistical analysis

Descriptive statistics were used to show the demographic and pathological characteristics of the patients. Pearson's chi-square or Fisher's exact test was used to analyze categorical variables, and the Mann–Whitney $U$ test was used to analyze ordinal variables. Multinomial logistic regression analyses were used to analyze associations between method of detection and pathological characteristics. The tumor size (T1, T2, and T3–4), node lymph stage (N0, N1, and N2–3), and histologic grade (I, II, and III) were treated as the outcome variables. The category "T1," "N0," and "I" was used as the reference category in tumor size model, node lymph stage model and histologic grade model respectively. The variable "age" was divided into four categories: cases younger than 40, 40–49, 50–59, and aged 60 and older. The variable "method of detection" and "age" was included in these models. The null hypothesis was that there would be no significant difference between variables. A significance level of 0.05 was used for two-tailed $P$ values.

## Techniques of dealing with missing data

To maximize the likelihood of comparability and equivalence, four methods were used to deal with missing data based on a missing-at-random assumption. These were as follows: (A) multiple imputation by chained equation (five times) (by R Project, version 3.3.2) (*Zhang, 2016*; *Eisemann, Waldmann & Katalinic, 2011*), with age group, T stage, N stage, histologic grade, and detection modes included into multivariate regression model; (B) complete randomized imputation (five times), stratified by year (*Pedersen et al., 2017*); (C) arbitrarily replacing all missing values for the detection methods into the self-detected mode and deleting other missing values in the group; (D) arbitrarily replacing a missing mode of detection into the screen-detected mode and deleting other missing values in the group.

**Table 1 Characteristics of the patients with breast cancer in Tianjin Medical University Cancer Institute and Hospital.**

| Characteristics | 1995–2000 ($n$ = 1,060) | 2010 ($n$ = 946) | 2015 ($n$ = 1,008) |
|---|---|---|---|
| Median age, years (range) | 48.0 (19–80) | 51.0 (22–82) | 52.0 (18–82) |
| Detection mode, $n$ (%) | | | |
| Self-detected | 1,034 (97.5) | 712 (75.3) | 774 (76.8) |
| Screen-detected | 4 (0.4) | 60 (6.3) | 76 (7.5) |
| Unknown | 22 (2.1) | 174 (18.4) | 158 (15.7) |
| T, $n$ (%) | | | |
| T1 | 168 (15.8) | 330 (34.8) | 299 (29.7) |
| T2 | 633 (59.7) | 352 (37.2) | 430 (42.6) |
| T3 | 127 (12.0) | 45 (4.8) | 46 (4.6) |
| T4 | 46 (4.3) | 13 (1.4) | 5 (0.5) |
| Unknown | 86 (8.2) | 206 (21.8) | 228 (22.6) |
| N, $n$ (%) | | | |
| N0 | 467 (44.1) | 390 (41.2) | 467 (46.4) |
| N1 | 397 (37.5) | 184 (19.5) | 243 (24.1) |
| N2 | 112 (10.6) | 104 (11.0) | 90 (8.9) |
| N3 | 4 (0.4) | 82 (8.7) | 124 (12.3) |
| Unknown | 80 (7.4) | 186 (19.6) | 84 (8.3) |
| Histologic grade, $n$ (%) | | | |
| I | 147 (13.9) | 103 (10.9) | 56 (5.5) |
| II | 605 (57.1) | 495 (52.3) | 684 (67.9) |
| III | 241 (22.7) | 86 (9.1) | 102 (10.1) |
| Unknown | 67 (6.3) | 262 (27.7) | 166 (16.5) |

**Notes:**
T, tumor size staging; N, lymph node staging.

# RESULTS

## Pathological characteristics of breast cancer patients

We identified 1,086, 1,053, and 1,047 female cases from databases in 1995–2000, 2010, and 2015, respectively. From these, we excluded 172 women with ductal carcinoma in situ or bilateral breast cancer. The final study therefore included 3,014 cases of invasive breast cancer: 1,060 in 1995–2000, 946 in 2010, and 1,008 in 2015. The median (range) ages at presentation were 48.0 (19–80) years in 1995–2000, 51.0 (22–82) years in 2010, and 52.0 (18–82) years in 2015. The general pathological characteristics of the cancers, including T stage, N stage, and histologic grade, are shown in Table 1 for each period.

## Pathological differences between the self-detected and screen-detected groups

The screen-detected group had a higher proportion of stage T1 tumors than the self-detected group in each database (75.0% vs 17.1%, $P$ = 0.109 in 1995–2000; 66.7% vs 40.4%, $P$ < 0.001 in 2010; and 67.8% vs 35.7%, $P$ < 0.001 in 2015) (Table 2; Fig. 1A). The proportion with negative lymph nodes (N0) was also slightly higher in the screen-detected

**Table 2 Comparison of differences in T stage, N stage, and histologic grade between self-detected and screen-detected patients.**

| Characteristics | | 1995–2000 | | 2010 | | 2015 | |
|---|---|---|---|---|---|---|---|
| | | Self-detected | Screen-detected | Self-detected | Screen-detected | Self-detected | Screen-detected |
| T stage, n (%) | T1 | 163 (17.1) | 3 (75.0) | 231 (40.4) | 34 (66.7) | 219 (35.7) | 40 (67.8) |
| | T2 | 620 (65.0) | –[a] | 289 (50.5) | 15 (29.4) | 350 (57.1) | 17 (28.8) |
| | T3–4 | 173 (17.9) | 1 (25.0) | 52 (9.1) | 2 (3.9) | 44 (7.2) | 2 (3.4) |
| *Mann–Whitney U* | | | 1,113.000 | | 10,651.500 | | 12,256.000 |
| *P* | | | 0.109[b] | | <0.001 | | <0.001 |
| N stage, n (%) | N0 | 455 (47.4) | 2 (50.0) | 284 (48.4) | 35 (67.3) | 347 (48.5) | 37 (55.2) |
| | N1 | 389 (40.6) | 2 (50.0) | 148 (25.2) | 10 (19.2) | 199 (27.8) | 16 (23.9) |
| | N2–3 | 115 (12.0) | –[a] | 155 (26.4) | 7 (13.5) | 170 (23.7) | 14 (20.9) |
| *Mann–Whitney U* | | | 1,754.000 | | 12,116.500 | | 22,349.000 |
| *P* | | | 0.868[b] | | 0.007 | | 0.325 |
| Histologic grade, n (%) | I | 143 (14.7) | 3 (75.0) | 72 (13.9) | 11 (22.9) | 38 (5.9) | 3 (4.6) |
| | II | 591 (60.8) | –[a] | 372 (71.8) | 35 (72.9) | 527 (81.6) | 57 (87.7) |
| | III | 238 (24.5) | 1 (25.0) | 74 (14.3) | 2 (4.2) | 81 (12.5) | 5 (7.7) |
| *Mann–Whitney U* | | | 1,067.500 | | 10,388.000 | | 20,270.000 |
| *P* | | | 0.100[b] | | 0.017 | | 0.491 |
| Age | <40 | 138 (13.3) | 2 (50.0) | 87 (12.2) | 3 (5.0) | 78 (10.1) | 3 (3.9) |
| | 40–49 | 428 (41.4) | –[a] | 218 (30.6) | 28 (46.7) | 233 (30.1) | 24 (31.6) |
| | 50–59 | 259 (25.0) | 2 (50.0) | 257 (36.1) | 20 (33.3) | 390 (37.5) | 27 (35.6) |
| | ≥60 | 209 (20.2) | –[a] | 150 (21.1) | 9 (15.0) | 173 (22.4) | 22 (28.9) |
| *Mann–Whitney U* | | | 1,529.000 | | 20,021.500 | | 26,602.000 |
| *P* | | | 0.400[b] | | 0.398 | | 0.149 |

Notes:
[a] There was no data in the current group.
[b] The P value was calculated by Fisher's exact test because the number of patients in the current group was less than five.

group than in the self-detected group in each database (50.0% vs 47.4%, 67.3% vs 48.4%, and 55.2% vs 48.5% in 1995–2000, 2010, and 2015, respectively), although statistical significance was only reached for 2010 ($P = 0.007$) (Table 2; Fig. 1B). The percentages of histologic grade I tumors were significant higher in screen-detected group than that in self-detected group (22.9% vs 13.9%, $P = 0.017$ in 2010) (Table 2; Fig. 1C). The age distribution showed no significant difference between self-detected and screen-detected group (Table 2; Fig. 2). After adjusting for age, self-detected group had increased T2 stage cases in 2010 (T2 vs T1, OR = 2.817, $P = 0.001$) and 2015 (T2 vs T1, OR = 3.820, $P < 0.001$), increased N2–3 stage cases in 2010 (N2–3 vs N0, OR = 2.775, $P = 0.017$) and increased histologic grade III in 2010 (grade III vs I, OR = 5.763, $P = 0.026$) significantly, compared with screen-detected group (Table 3).

## DISCUSSION

In this study, we retrospectively compared the differences in pathological characteristics between self-detected and screen-detected breast cancer. The proportion of cases identified by the screening program increased significantly before the introduction of screening.

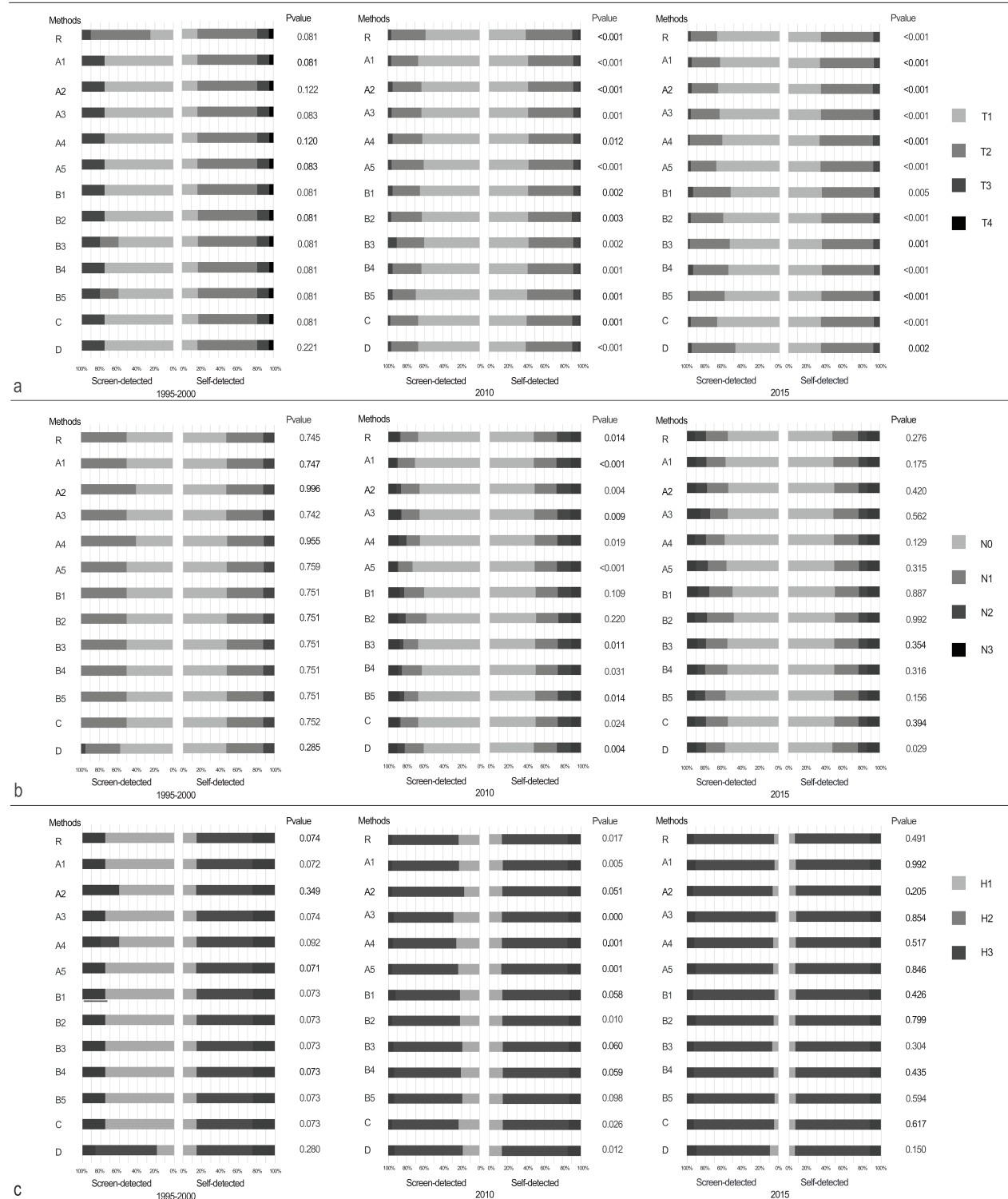

**Figure 1 Comparison of the difference between self-detected and screen-detected breast cancer patients in (A) T stage, (B) N stage, and (C) histologic grade in 1995–2000, 2010, and 2015.** Techniques of dealing with missing data included (R) complete-case analysis; (A1–5) multiple imputation by chained equations; (B1–5) completely randomized imputation; (C) arbitrarily replacing missing mode of detection into self-detected mode and deleting other missing values in the group; (D) arbitrarily replacing all missing detection method values into screen-detected mode and deleting other missing values in the group.               
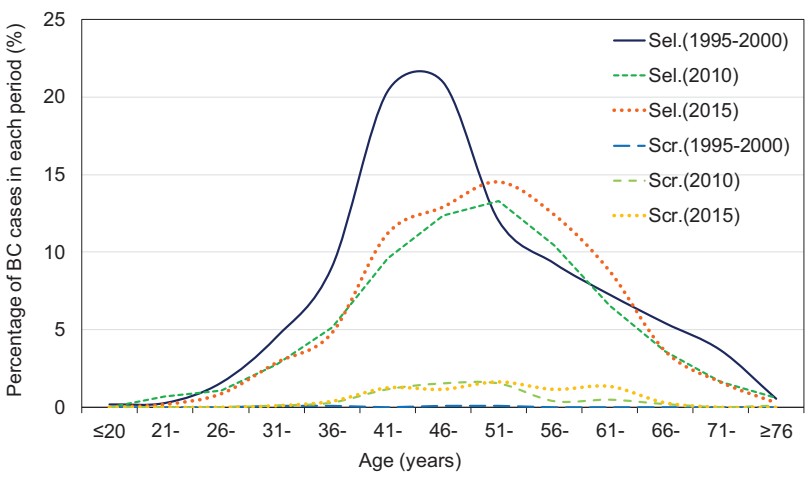

**Figure 2 Frequency distribution at diagnosis, by detection mode in 1995–2000, 2010, and 2015.**
The age distribution of screen- and self-detected patients was constructed using the 2016 Excel software, while the patients with missing values of detection mode were deleted. Periods of self-detected patients included 1995–2000 in full line (Sel.1995–2000), 2010 in dotted dot line (Sel.2010), and 2015 in square dot line (Sel.2015). Periods of screen-detected patients included 1995–2000 in dash-dot line (Scr.1995–2000), 2010 in long dashed line (Scr.2010), and 2015 in short dashed line (Scr.2015).

**Table 3 Relationship between pathological characteristics and method of detection of breast cancer patients after adjusting for age.**

| Characteristics | | 1995–2000 | | | 2010 | | | 2015 | | |
|---|---|---|---|---|---|---|---|---|---|---|
| | | OR[a] | 95% CI | P | OR[a] | 95% CI | P | OR[a] | 95% CI | P |
| T stage | T2 vs T1 | –[b] | –[b] | –[b] | 2.817 | 1.497–5.300 | 0.001[c] | 3.820 | 2.111–6.915 | <0.001[c] |
| | T3–4 vs T1 | 2.961 | 0.303–28.926 | 0.351 | 0.023 | 0.888–16.397 | 0.072 | 3.835 | 0.891–16.498 | 0.071 |
| N stage | N1 vs N0 | 0.851 | 0.119–6.078 | 0.872 | 1.832 | 0.882–3.806 | 0.105 | 1.339 | 0.726–2.469 | 0.351 |
| | N2–3 vs N0 | –[b] | –[b] | –[b] | 2.775 | 1.203–6.400 | 0.017[c] | 1.308 | 0.688–2.486 | 0.413 |
| Histologic grade | II vs I | –[b] | –[b] | –[b] | 1.636 | 0.793–3.375 | 0.182 | 0.725 | 0.217–2.424 | 0.601 |
| | III vs I | 4.544 | 0.463–44.572 | 0.194 | 5.763 | 1.233–26.945 | 0.026[c] | 1.251 | 0.284–5.517 | 0.767 |

Notes:
[a] OR, odds ratio values. The OR value and P value was calculated by using a multinomial logistic regression model after adjusting for age.
[b] There was no data in the current group.
[c] P value indicates statistical significance at the 0.05 level.

The screen-detected group had smaller tumor sizes and tended to have less lymph node involvement and lower histologic grades compared with the self-detected group.

The coverage of the breast cancer screening remains low in Chinese population. From 2009 to 2011, a breast cancer screening program, which was launched by the Chinese Anti-Cancer Association with the permission of the Chinese government, only covered 1.46 million women and only 631 with breast cancer (*Song et al., 2015*). As of 2014, the total number of screened women had risen to 48.35 million, but this still accounts for less than 5% of the population. Another possible explanation for the low percentage of screen-detected cancer may relate to the theory and technology underpinning existing screening programs and guidelines, typically relying on a lack of indigenous studies. Moreover, the Chinese guidelines for breast cancer screening were not

published by the Chinese Anti-Cancer Association, Committee of Breast Cancer Society until 2007 (*Chinese Anticancer Association Breast Cancer Society, 2007*) and have been updated four times over the last decade. These guidelines recommend that women at average risk of breast cancer only undergo opportunistic screening mammography. However, ultrasound and parallel clinical breast examination are the primary screening tools in second-generation screening programs (*Song et al., 2015*).

Consistent with the findings of previous studies from Japan, Singapore, Korea, and some Western countries, we confirmed the benefits of screening when seeking to detect breast cancer at an early stage (*Joensuu et al., 2004*; *Kim et al., 2012*; *Crispo et al., 2013*; *Inari et al., 2017*; *Chuwa et al., 2009*). Specifically, we identified the prognostic advantages, based on pathological findings at diagnosis, for asymptomatic patients with screen-detected cancers. Comparable to our results (66.7–75.0% vs 17.1–40.4%), higher proportions of screen-detected patients were reported to have stage T1 cancer compared with self-detected groups in studies in both Korea (59.2% vs 31.7%) (*Kim et al., 2012*) and Hong Kong (44.7% vs 33.4%) (*Lau et al., 2016*). A study in Singapore also indicated that screening was an independent factor for better clinical staging at presentation, after adjusting for race and menopausal status (*Wang, Tan & Chow, 2011*). However, although there were trends, we did not find any statistically significant difference for lymph node status or histologic grade between the groups in this study. Mammography was not popular yet as the modality of breast cancer screening in China (*Song et al., 2015*; *Chinese Anticancer Association Breast Cancer Society, 2015*). Breast cancer examination was used in most of the screening program which might limit the performance of screening and results unsatisfied tumor stage.

In this study, long-term information of 3,014 breast cancer patients from Tianjin Medical University Cancer Institute and Hospital were collected. Because the breast cancer patients at our hospital came from all over the country of China, this database represent a trend of Chinese breast cancer. However, this study has two main limitations. The first is that it was retrospective and that approximately 12% of values were missing in the detection mode due to the use of electronic documentation. Hence, four imputation methods were used to ascertain whether major differences occurred on the comparison of pathological characteristics between self-detected and screen-detected breast cancer. When using multiple imputation by chained equations, the missing values were completed depending on the interdependency between values (*Zhang, 2016*). In this regard, more preferable results tended to be classified into the screen-detected group. When using completely randomized imputation stratified by year, no tendency was seen in either direction. When the missing detection mode values were replaced by "self-detected," the pathological advantage of the screen-detected group was attributed to the self-detected group. The differences between the two groups may therefore have been underestimated. When the missing detection mode values were replaced by "screen-detected," the disadvantage in the self-detected group was attributed to the screen-detected group, also potentially leading to an underestimation of the differences between the two groups. The second limitation is that there was no information about the follow-up or survival status of the patients, for which further studies would be

required. A study from the UK reported that the impact of introducing such a screening program on survival was small but significant, and that most of the improved survival was due to a shift in the Nottingham Prognostic Index (used to determine prognosis following surgery for breast cancer) (*Wishart et al., 2016*). Similar shifts in pathologic characteristics of prognosis were identified both in this retrospective investigation and in previous studies (*Zheng et al., 2012*).

## CONCLUSION

This study indicates that the breast cancers detected by screening had more favorable clinicopathologic characteristics than those detected by themselves. We also found fewer patients were detected by screening in this study compared with those in Western and Asian developed countries.

## ACKNOWLEDGEMENTS

We thank Dr. Robert Sykes for providing editorial services.

### Funding
The work was supported by grants from the National Natural Science Foundation of China (No. 81301799 and No. 81172531). The funders had no role in study design, data collection and analysis, decision to publish, or preparation of the manuscript.

### Grant Disclosures
The following grant information was disclosed by the authors:
National Natural Science Foundation of China: 81301799 and 81172531.

### Competing Interests
The authors declare that they have no competing interests.

### Author Contributions
- Qi Zhang conceived and designed the experiments, performed the experiments, approved the final draft.
- Lanjun Ding conceived and designed the experiments, analyzed the data, prepared figures and/or tables, approved the final draft.
- Xuan Liang approved the final draft, help to analysis.
- Yuan Wang contributed reagents/materials/analysis tools, approved the final draft.
- Jiao Jiao approved the final draft, helped to collect Casea and revised article.
- Wenli Lu authored or reviewed drafts of the paper, approved the final draft.
- Xiaojing Guo authored or reviewed drafts of the paper, approved the final draft.

### Human Ethics
The study was approved by the Ethics Committee in Tianjin Medical University Cancer Institute and Hospital. The ethics committee waived the need for informed consent.

## Data Availability

The raw data have been provided as a Supplemental Dataset.

## Supplemental Information

Supplemental information for this article can be found online at http://dx.doi.org/10.7717/peerj.4567#supplemental-information.

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
