# Peer review of "Comparison of pathological characteristics between self-detected and screen-detected invasive breast cancers in Chinese women: a retrospective study"

_PeerJ, doi:10.7717/peerj.4567_

## Round 0.1 · original submission · Major Revisions

· Academic Editor

Major Revisions

Please provide a point-to-point response in your revision. A more careful review of the grammar and writing style is recommended.

Reviewer 1 ·

Basic reporting

1. This study use the China breast screen program as a natural experiment to explore the prognostic characteristics pre and post, and between those self-detected and screen-detected. In the introduction section (second paragraph), please clarify whether there are similar studies done or not and if not, how this study fill in the literature gap.
2. The English language is still a little awkward and confusing in a few places. One example is that in the result section (Line 147), the word “comparable” means similar/equivalent whereas the P-value is actually <0.05. The sentence in the method section (line 93-94) also looks awkward.
3. The abstract “Methods” section should also contain description of your statistical analysis approach. The P-value format should also be corrected (i.e. 0.000 should be <0.001).

Experimental design

1. This retrospective study used patient medical records from three different periods, which derived from different sources (paper documentation and electronic documentation). Please clarify whether there is quality control process of abstracting the data. In addition, were the missing data due to the use of electronic documentation? The records from the latter two periods (1995-2000, and 2015) showed higher % of missing data which happened to be the two periods that used electronic documentation. Or was there a systematic reason that caused the missing data?
2. There are a few issues with the author’s statistical analysis approaches. First, raw cases of screen-detected patients and self-detected patients should not be compared directly. The increases of screen-detected cases might be due to more women being screened in 2015, and fewer women with symptoms visited the hospital in 2015. The other similar studies cited in the manuscript (Ref [13], [22], etc.) did not compare the raw cases. I suggest taking out comparison from the second paragraph in the result section (line 130-139), as well as in the discussion section (line 156-167).
3. Multiple Imputations: this approach is based on multivariate regression. I am concerned about the validity of the regression because there are very few variables provided in the study. Please clarify what variables you put into the model.
4. Unadjusted comparison of prognostic factors between self-detected and screen-detected is not sufficient. The other Asian studies cited in the manuscript all showed significantly different patient characteristics between the self-detected group and screen-detected group, and some of these characteristics may also affect prognostic factors. Although very few patient characteristics were available in this study, the author could still try to adjust for age or output age-standardized distribution of prognostic factors. As shown in Table 2, the age range is big and considerably bigger in self-detected group (compared to screen-detected group) especially in earlier period.

Validity of the findings

1. As mentioned above, the raw cases of self-detected and screen-detected should not be compared directly and therefore the discussion (line 156-157) is not valid.
2. This study used data from Tianjing Medical University Cancer Institute and Hospital, which is only a small subset of the national data. Given the huge regional variation of cancer incident rate in China, the screen-detected from this study may not be representative of China. Therefore, the conclusion (line 210) that “this study indicates that the proportion of screen-detected patients in China remains lower than that in other Asian developed countries and regions” is not valid. The results from this study cannot be generalized to the whole country.

Additional comments

1. Please revise or take out the comparison of raw number between self-detected and screen-detected cases.
2. I suggest adding in and controlling for some patient characteristics if you have them available.
3. Please review the paper and make sure the format of the numbers (i.e. P-value) is correct.
4. Please carefully refine the conclusion generated from your results.

·

Basic reporting

1. In this manuscript mentioned there are studies worldwide have indicated that screen-detected patients have more favorable survival outcomes compared with patients that present with self-discovered breast cancer symptoms. Western and some Asian developed countries include Korean, Japan, Singapore. Please emphasize that this study use long term and huge database. Because used total sample were 3,014 female patients with invasive breast cancer. Describe and emphasize these database represent a trend of Chinese breast cancer.
2. It is probably more appropriate use the term of “characteristics of the cancers detected by screening and those by self-exam.” The term “prognostic factors” is more appropriate for comparing survivals while the factors too are prognostic factors.
3. Line 33, “changed” seems to be less desired than “differed.” There are many more grammatical errors and writing style issues.
4. Figure 2, the lines are difficult to differentiate between the groups. Please change them to color-lines or other format.

Experimental design

1. The study design is not very clear to me. If it were to examine the effects of the screening-policy implementation, the authors ought to consider the factors associated with the implementation as this is an intervention-based cohort study. If they wanted to compare the cancers detected by self-exam and those by screening, then what study design did they have?
2. The 1995-2000 had too few (absolute number) screening-detected cases. The lack of statistical significance (P=.08) may be associated with the small sample size. In fact, because n=4, Fischer exact test should have been used.
3. What are the factors associated with screening for breast cancer? Those factors may have driven or linked to the differences shown in the study.

Validity of the findings

In this manuscript, Zhang Qi et al reported comparison of prognostic factors between self-detected and screen-detected invasive breast cancers in Tianjin, China.
They compared the prognostic factors at diagnosis between self-detected and screen-detected cases of invasive breast cancer in Tianjin, use three databases were selected breast cancer diagnosed in 1995–2000, 2010, and 2015 groups. Data include patient’s clinical history and prognostic factors, plus the cancer detection methods.
They found Screen-detected breast cancer was associated with favorable prognostic factors.
The results in this study, confirmed the benefit of early detection through a screening program for breast cancer in Tianjin, China.

Additional comments

This study is well written, there are comments to change before acceptable for publication.

---

## Round 0.2 · Minor Revisions

· Academic Editor

Minor Revisions

See the remaining comments from Reviewer 1 and some small changes recommended by me in the attached pdf file.

Thank you!

Reviewer 1 ·

Basic reporting

All prior comments were addressed. Good job!

Experimental design

Most comments were addressed appropriately, except for the final multivariate regression analysis. It looks like the current regression use T stage, N stage, and histologic grade as continuous because only one Odd Ratio was provided for each variable. However, these three variable should be used as categorical variable and there should be one Odd Ratio for each category except for the reference group. Since there are small cells for certain categories, you can consider combining some of the categories into bigger group. If the distribution of age is not normal or extreme, can also consider categorizing it into several groups as well.

Validity of the findings

All comments addressed appropriately.

Additional comments

The revision is overall good. But need to make some adjustment to the multivariate logistic regression (using T stage, N stage, and historic grade as categorical variable) and modify the corresponding "Result" and "Discussion" part accordingly.

·

Basic reporting

no comment

Experimental design

no comment

Validity of the findings

no comment

Additional comments

The authors have been answered all the questions and made the appropriate modifications.
I agree to accept your manuscript.
Thank you.

---

## Round 0.3 · accepted · Accept

· Academic Editor

Accept

Please include the recommended minor changes (see attached) in the abstract during copy editing. It will save time in processing.